# DNA Damage Response Inhibitors in Cholangiocarcinoma: Current Progress and Perspectives

**DOI:** 10.3390/cells11091463

**Published:** 2022-04-26

**Authors:** Öykü Gönül Geyik, Giulia Anichini, Engin Ulukaya, Fabio Marra, Chiara Raggi

**Affiliations:** 1Department of Medical Biology, Faculty of Medicine, Istinye University, Istanbul 34010, Turkey; oyku.geyik@istinye.edu.tr; 2Molecular Cancer Research Laboratory (ISUMCRC), Istinye University, Istanbul 34010, Turkey; eulukaya@istinye.edu.tr; 3Core Research Laboratory, Institute for Cancer Research and Prevention (ISPRO), 50139 Florence, Italy; giulia.anichini@gmail.com; 4Department of Medical Biochemistry, Faculty of Medicine, Istinye University, Istanbul 34010, Turkey; 5Department of Experimental and Clinical Medicine, University of Florence, 50139 Florence, Italy

**Keywords:** biliary tract cancer, targeted therapy, DNA damage, synthetic lethality, PARP, Wee1

## Abstract

Cholangiocarcinoma (CCA) is a poorly treatable type of cancer and its incidence is dramatically increasing. The lack of understanding of the biology of this tumor has slowed down the identification of novel targets and the development of effective treatments. Based on next generation sequencing profiling, alterations in DNA damage response (DDR)-related genes are paving the way for DDR-targeting strategies in CCA. Based on the notion of synthetic lethality, several DDR-inhibitors (DDRi) have been developed with the aim of accumulating enough DNA damage to induce cell death in tumor cells. Observing that DDRi alone could be insufficient for clinical use in CCA patients, the combination of DNA-damaging regimens with targeted approaches has started to be considered, as evidenced by many emerging clinical trials. Hence, novel therapeutic strategies combining DDRi with patient-specific targeted drugs could be the next level for treating cholangiocarcinoma.

## 1. Introduction

Most tumor hallmarks described by Hanahan and Weinberg [1,2,3] are directly associated with DNA damage-related alterations in cancer cells. Indeed, the “genome instability and mutation” hallmark is not only one of the leading forces of carcinogenesis, but is also a therapeutic target for inhibition of DNA damage response (DDR) elements such as PARP, ATM, ATR, etc. By increasing genomic instability, DDR inhibitors provide the opportunity to force tumor cells over the edge into apoptosis.

Primary liver cancers, including cholangiocarcinoma (CCA), are the third leading cause of cancer mortality worldwide [4]. In 2020, out of approximately 900,000 patients diagnosed, 830,000 have died. According to a recent report examining twenty-year data on global incidence and trends, both intrahepatic (iCCA) and extrahepatic (eCCA) incidence increased worldwide [5] and this is estimated to continue to grow due to metabolic and infectious etiologic factors in the next years.

Although some advances have been accomplished in the therapeutic approaches of CCA over the years, the survival rate is still not satisfying. The first choice of CCA therapy is resection, and adjuvant treatment is used for unresectable cases [6,7]. If the tumor can be removed successfully, patients can also be administered adjuvant chemotherapy or radiotherapy post-operation to increase complete recovery chances [8,9]. The most common chemotherapeutics used in CCA treatment include: 5-Fluorouracil with Folinic acid [10], Gemcitabine [11], Gemcitabine with Cisplatin [12], Irinotecan [13], or Capecitabine [14]. Interestingly, it has been shown that the tyrosine kinase inhibitor Erlotinib is beneficial in advanced CCA [15], and importantly, radiotherapy ameliorates the survival rate of CCA patients [16,17]. Local treatments are also currently used for CCA including transarterial chemoembolization (TACE), selective internal radiation therapy (SIRT), radiofrequency ablation (RFA), photodynamic therapy (PDT), and brachytherapy (iodine-125 seed implantation) [18,19], thus providing a beneficial extension of survival time. Nevertheless, an increase in survival and quality of life can be achieved by moving towards a more personalized medical approach [20,21]. Although several oncogenic pathways have been identified in CCA, curative therapies have been difficult to develop due to the extreme genetic heterogeneity and drug resistance [9,22,23,24].

Therapy resistance related to tumor heterogeneity is mostly attributed to genetic instability. Since the link between chromosomal abnormalities and cancer was first proposed [25], accumulating evidence has associated numerical and structural chromosomal aberrations to aggressive tumor behavior [26,27,28]. It has been assumed that genetic instability is a fundamental feature of cancer [29] and recent studies have strengthened the suggestion that instability-conferring mutational changes occur early during tumorigenesis [30,31,32,33]. Subsequent genetic instability generates mutations in proto-oncogenes and tumor suppressor genes, triggering tumor progression. The prevalence of genomic instability points to multiple cancer-associated pathways, whose deregulation has been implicated in affecting mitotic chromosome segregation [34,35,36,37].

Genomic instability has long been thought to facilitate treatment resistance by causing heterogeneity at the gene level. However, to maintain the balance between genomic chaos and the acquisition of heterogeneity, cancer cells must constrain chromosome mis-segregation rates within a limited range that maximizes their viability [38,39]. The anti-neoplastic effects of therapies that promote chromosomal instability rest on this premise. For example, widely used anticancer agents such as Taxol, PARP inhibitors and ionizing radiation (IR) are among the strongest inducers of chromosome segregation errors [40,41]. Specifically, exposure to IR promotes the generation of a variety of different lesions within DNA architecture. Radiotherapy-induced DNA damage arises following direct ionization on DNA sugar backbone, or indirectly, by promoting the production of free radicals in cells, which yield oxidative DNA damage and replication stress [42,43]. IR-induced DNA damage includes base modifications (8-OxoG), in addition to crosslinks and DNA ruptures, both at single strand break (SSB) and double strand break (DSB) levels, progressively promoting genomic instability and the activation of DNA damage response patterns in cancer cells [44,45]. DDR activation supports DNA repair and the development of radio-resistance, contributing to therapy failure. Radiotherapy is currently indicated for different CCA patients both in adjuvant regimens and for unresectable. Hence, it is considered rational to target this genetic heterogeneity with DDR inhibitors [46,47,48,49,50,51,52,53]. Using these beneficial tools in combination with patient-specific targeted drugs holds great promise to overcome CCA [54,55,56].

In CCA, DDR inhibitors have a broad area of application likely due to the high aggressivity associated with increased genomic instability. Indeed, both chromosomal or sequence-specific variability represent a fundamental cancer feature that is associated with poor prognosis, metastasis, and therapeutic resistance. It results from errors in chromosome segregation and cell cycle checkpoints during mitosis and leads to both structural and numerical abnormalities. Additionally, genomic instability regulates immune escape by modulating the interaction between tumor and its microenvironment thus driving tumor growth.

In this review we discussed the use of DDR inhibitors in CCA and the combinations of them with other chemotherapeutic or targeted agents (i.e., immunotherapeutic and antifibrotic drugs).

## 2. Targets and Related Therapies for DDR in Cholangiocarcinoma

Development and progression of CCA have been widely associated with increased DNA damage and genomic instability. Next-generation profiling of CCA as part of the Pan Cancer Analysis of Whole Genome initiative under the management of The International Cancer Genome Consortium (ICGC) project [57,58,59] and United States National Cancer Institutes’ the Cancer Genome Atlas (TCGA) which investigates the genomic alterations in more than 50 different cancers including BTCs [60], revealed the presence of alterations in DDR-related genes, including BRCA1/2, PARP, ATM, ATR, BAP1, ARID1A, RAD51, MLH1, TP53, PALB2, PTEN, FANC, NBN, EMSY and MRE11 [61,62,63,64,65,66,67,68], with the most commonly altered DDR genes being ATM (5%) and BRCA1/2 (4.8%) [62]. According to a very recent study on the genetic determinants of iCCA, the most common oncogenic alterations were IDH1 (20%), ARID1A (20%), TP53 (17%), CDKN2A (15%), BAP1 (15%), FGFR2 (15%), PBRM1 (12%), and KRAS (10%). IDH1/2 mutations were mutually exclusive with FGFR2 fusions (fus) and TP53 (*p* < 0.0001), KRAS (*p* = 0.0001), and CDKN2A (*p* < 0.0001) alterations predicted worse overall survival (OS) [69]. DDR dysregulation is currently considered one of the most relevant intracellular pathways in regulating proliferation, apoptosis and chemoresistance in biliary tract cancers (BTC) [64,70], as germline or somatic mutations in DDR genes were found in the majority of patients (63.5% [67]) and the presence of mutations/alterations in the expression of specific DDR genes can impact on patient response to platinum-based regimens [67,71,72,73,74]. Based on the well-established concept of synthetic lethality (SL), where mutation of two specific genes individually is compatible with viability, but mutation of both leads to cell death, several DDR targeting strategies have been developed [75]. The idea is that targeting one DDR gene in cancer cells that are defective for other DDR genes might be useful, in order to acquire enough DNA damage to trigger cancer cell death [63,76,77]. Thus, targeting DDR genes that are synthetically lethal with a mutation that is common in cancer, can selectively eliminate cancer cells without harming healthy cells. SL, therefore, provides an important means of strategy for the development of cancer-specific therapies [78]. Moreover, pharmacological targeting of DDR strengthens the cytotoxic effects of DNA damaging chemotherapeutics, which still unfortunately represent the main therapeutic regimen for CCA patients, wherein the development of chemoresistance drastically affects the therapeutic outcomes. Hence, different DDR inhibiting molecules are under evaluation both at the preclinical and clinical level for different types of human cancers, including CCA.

### 2.1. PARP

The PARP (poly (ADP-ribose) polymerase) family of enzymes includes 17 proteins, with PARP1 accounting for 80% of DDR activity [79,80]. PARP1, together with PARP2, function as a DNA damage sensor and signal transducer by recognizing damaged sites in the DNA structure. Upon its binding to DNA breaks, PARP synthetizes negatively charged PAR chains, which are, therefore, attached to target proteins through the PARylation process, a mechanism involved in modifying the chromatin architecture and supporting the recruitment of repair proteins to the sites of damage [81,82]. Once the DNA repair process restores DNA integrity, PARP1 PARylates itself and is released from DNA ruptures [77,83,84]. Interestingly, PARP enzymes participate in regulating not only SSB repair, but also DSB DNA repair through homologous recombination repair (HRR) and non-homologous end joining (NHEJ) mechanisms [85]. PARP1 inhibition leads to persistent SSBs that might result in the disruption of the replication fork, thus generating DSBs [86,87]. In addition, anti-PARP agents preclude autoPARylation and forcefully hold PARP1 on damaged DNA, hampering repair processes and leading to cell death [82,88,89]. Finally, upon the inhibition of PARP-mediated DNA repair, cancer cells are forced to compensate with alternative repair mechanisms, such as NHR, which are more subjected to the introduction of errors and might culminate with massive genomic rearrangements and apoptotic cell death [77,90,91,92]. Hence, using orally deliverable small-molecule inhibitors which target PARP1, PARP2, and PARP3 (PARPi) results in strong anti-cancer effects and to date, several PARP-targeting compounds have been developed. Olaparib (Lynparza) is the first PARPi to be FDA-approved for the management of BRCA mutated ovarian, breast, pancreatic and prostate cancers [93,94,95,96,97,98,99]. Following Olaparib, several alternative PARP inhibiting molecules have been tested in both preclinical and clinical studies, namely, Niraparib, Talazoparib, Rucaparib, Veliparib, all showing overall favorable safety profiles [100,101,102].

Active clinical studies of DDR inhibitors in CCA patients were given in Table 1. To date, disparate PARP inhibiting compounds are under evaluation in CCA, both in preclinical and clinical settings. Preclinical studies have provided promising data about the efficacy of the pharmacological inhibition of PARP in CCA cells, alone or in combination with standard chemotherapy and/or targeted molecules [66,103,104,105]. Current data about the use of PARPi in BTC patients characterized by DDR dysregulation are still open to question, with OS profiles covering from 11 to 65 months [95,103,106,107]. Specifically, CCA patients are presently enrolled in disparate phase I-II trials for the clinical evaluation of the effectiveness of PARPi as anti-CCA agents for monotherapeutic or combination-based approaches [108]. The FDA-approved PARPi Olaparib is under clinical assessment as monotherapy for the determination of overall response rate (ORR) in CCA patients with metastatic disease and alterations in DDR genes [109]. IDH1/2 genetic aberrations occur in 20% of iCCA cases and may be specifically targeted by IDH inhibitors [110]. IDH1/2 mutations have been demonstrated to increase the sensitivity to PARPi [111], therefore, patients with metastatic CCA, harboring IDH1/2 genetic alterations are currently enrolled for the evaluation of ORR and progression free survival (PFS) following Olaparib administration as single-agent therapy [112]. Partly due to the described onset of PARPi-resistance in oncological patients [113], multiple-targeting approaches by using different PARPi-based combinatorial regimens are also under clinical testing. Namely, patients with metastatic and refractory solid malignancies, including bile duct neoplasms, are currently under active recruitment for the evaluation of the therapeutic response to Olaparib administered in combination with other DDR targeting compounds [114,115] and immunotherapy [116,117]. This latter combinatorial approach has revealed, intriguingly, to be a novel potent strategy in the field of anti-tumor therapy, as dysregulated DDR appears to improve immune recognition of cancer cells, by boosting local neoantigen exposure and DAMPs release [118,119,120].

Beside Olaparib, clinical evaluations of other potent PARPi in CCA patients are ongoing, mainly in combinatorial plans with systemic therapies. The single-arm phase I study NCT03337087 aims to estimate tolerability and anti-disease effects of the recently FDA-authorized Rucaparib [107] in combination with different standard chemotherapeutics (Irinotecan, Fluorouracil, Leucovorin) in patients with gastrointestinal tumors, including CCA. Moreover, phase II studies are elucidating the effects of combining Rucaparib-mediated PARP inhibition and immunotherapeutic targeting of PD-1 in CCA patients who already completed a first-line platinum-based therapy for 4–6 months, without any clinical progression [121]. Improving outcomes for clinical PFS and overall survival (OS) are expected. Similar effects were supposed for the phase I evaluation of the potent orally bioavailable PARPi ABT-888 (Veliparib) in CCA patients treated with Gemcitabine and/or Cisplatin [122]. However, to date data are still not available. Finally, investigators plan to assess ORR, together with PFS and OS, following once-daily Niraparib administration in patients with refractory CCA. The recruiting phase recently concluded, and patients who had already received first-line standard therapy have been selected according to the presence of genetic alterations in DDR genes, including BAP1, ATM, ATR, CHK1/2, RAD51 and others [123].

### 2.2. Wee1

The Wee1 family of Ser/Thr protein kinases includes mainly three members: Wee1, Myt1 (PKMYT1) and Wee1B (Wee2). The first two are implicated in controlling the G2/M transition in somatic cells, being indispensable for regulating the CDK1-Cyclin B1 complex and thus entry and progression of cells into mitosis, whereas Wee2 is involved specifically in the biology of gamete cells, where it regulates the meiotic process [124,125]. Therefore, while Wee1 and Myt1 are expected to work as tumor suppressor genes, they appear to acquire oncogenic properties in malignant cells. In fact, Wee1 and Myt1 have been found overexpressed in a variety of tumors, and a large-scale CRISPR screening revealed that both kinases are essentially required for cancer cell viability [126]. Cancer cells might be dependent on Wee1 family kinases, as their high replication rate needs a powerful network in regulating cell cycle and as p53 is often inactivated in tumors, cancer cells usually rely entirely on the G2/M checkpoint. Moreover, maintaining a sustainable degree of genomic instability in malignant cells appears to require the overexpression of DDR-associated kinases [127,128,129].

Therefore, Wee1 family has been investigated as a potential therapeutic target for human tumors, with a variety of molecules being synthetized and evaluated for anti-cancer activity in several malignancies. Essentially, Wee1/Myt1 inhibiting agents, alone or in combination with chemo- and radiotherapy, function by inducing mitotic catastrophe, as they force G2 transition, also in presence of extensive damaged DNA, and premature entry into mitosis [129]. Until the present time, a plethora of different Wee1 targeting compounds have been preclinically screened and checked in different tumors, including CCA, with encouraging anti-cancer activity both as single agents and in combination with systemic or targeted therapeutics [129,130,131,132]. Among all, Adavosertib (AZD1775, MK1775) and IMP7068 have been chosen for further clinical applications. The orally deliverable small-molecule Adavosertib essentially represents the predominantly studied Wee1 inhibitor, alone or in combinatorial regimens with diverse therapeutics. Intriguingly, phase I-II studies in different malignancies, including CCA, revealed satisfactory outcomes about overall safety and tolerability, along with improving ORR and PFS rates [129,133,134,135,136,137], suggesting that, together with PARP, Wee1 might represent a prominent target to develop in setting up novel therapeutic programs for CCA patients. Interestingly, CCA patients are currently enrolled in the phase II MATCH clinical trial [138], which is actively recruiting patients with refractory BRCA1/2 mutated intrahepatic CCA, to further corroborate data about patients ORR and OS following Adavosertib administration.

### 2.3. ATR and ATM

In mammalian cells, ATR (ataxia telangiectasia and Rad3-related protein kinase) and ATM (ataxia-telangiectasia mutated protein kinase) represent the main transducers of DNA damage response pathways, coordinating a complex network of cellular processes involved in preserving genomic stability. ATM is predominantly activated upon DSBs, while ATR is implicated in responding to a wider landscape of genotoxic insults [139]. DNA break-points are detected and occupied by replication protein A (RPA), which is then recognized by ATR-interacting protein (ATRIP). ATR kinase is then recruited to DNA breaks, where it is phosphorylated and activated, starting a downstream cascade which induces, among others, the activation of checkpoint kinase 1 (CHK1), a crucial mediator of cell cycle checkpoint regulation [140,141,142]. Conversely, ATM is known to be the prevalent regulator of DSB-induced DNA damage response. By interacting with Nbs1, ATM is recruited to DNA breaks and activated, thus phosphorylating several substrates, including the checkpoint kinase 2 (CHK2) and histone H2AX [143,144,145], which facilitates DNA repair by reducing chromatin density and by originating an epigenetic signal that is recognized by specific DDR proteins [146,147,148,149].

In contrast with PARPi and Wee1, clinical development and application not only of ATR and ATM inhibitors (ATRi, ATMi), but also of the other targets described below, are still quite preliminary. Interestingly, preclinical data suggest that the pharmacological inhibition of ATR and ATM, especially when combined with DNA-damaging agents or PARPi, offers a feasible therapeutic approach to CCA [150,151,152,153]. However, to our knowledge there are no clinical studies enrolling CCA patients for the evaluation of therapeutic efficacy of ATMi. Conversely, the ATP-competitive ATRi Berzosertib (M6620, VX-970) alone or in combination with chemotherapeutics was evaluated in patients with advanced solid malignancies, including CCA, and showed consistent biosafety and tolerability, in addition to providing positive anti-tumor rates in phase I studies [154,155]. Accordingly, the potent oral selective ATRi Ceralasertib (AZD7638), which showed potent antineoplastic properties in several both preclinical and clinical studies (reviewed in [156]), has been selected for clinical use in the treatment of CCA. Multiple-targeting phase II studies are currently evaluating therapeutic advantages of combined administration of Ceralasertib with PARPi for immunotherapy in patients with solid CCA and other malignancies [114,115] who have failed the first-line systemic therapy. As far as it is known, there is no active clinical evaluation of other ATRi in CCA patients, since the phase I evaluation of the ATR kinase inhibitor Elimusertib (BAY 1895344) in BTC and other advanced tumor patients has been suspended due to unacceptable toxicity [157].

### 2.4. CHK1 and CHK2

As mentioned before, cell cycle checkpoint kinases CHK1 and CHK2 are the major downstream targets of ATR and ATM kinases. Both ATR-CHK1 and ATM-CHK2 signaling cascades converge to the negative regulation of CDC25 family of phosphatases, which crucially manage the phosphorylation status of CDK-Cyclin complexes, being thus key regulators in controlling G1/S and G2/M transitions.

Pharmacological inhibition of CHK1 and CHK2 kinases produces forced entry into mitosis and results in accumulation of DNA damage, ending with cell death [158]. Several specific CHK1/2 inhibiting compounds have been developed thus far. Nevertheless, the application of these compounds in CCA is still quite early and basically poorly explored. A preclinical study suggests that CHK1/2 inhibition by AZD7762 sensitizes CCA cells to radiotherapeutic effects [159]. As far as we know, NCT02124148 is the sole clinical application involving metastatic CCA patients elucidating the efficacy of combining the dual CHK1/2 kinase inhibitor Prexasertib (LY2606368) with standard chemotherapeutics, including cisplatin, showing promising phase I indications about patient response and drug tolerability [160]. Even though available studies provide mainly preliminary information and wider research appears required, current data suggest that targeting DDR at the level of downstream mediators CHK1/2 might represent a reasonable strategy to deepen for the development of novel therapeutic programs for CCA patients.

### 2.5. Other Targets

As mentioned, several different putative mechanisms can be targeted with the aim of inhibiting DNA damage response pathways. In addition to the most studied proteins such as PARP, Wee1, CHK1/2, ATR and ATM, other druggable targets are currently under evaluation to further the development of molecules with a specific pharmacological activity against DDR, including DNA dependent protein-kinase (DNA-PK) and polo-like kinase 1 (PLK1).

#### 2.5.1. DNA-PK

DNA-PK is a Ser/Thr kinase which belongs to the phosphatidylinositol 3-kinase-related kinase (PIKK) family with a key function during DSBs repair, cell cycle progression and preservation of telomeres [161]. DNA-PK has been shown to be upregulated in BTC, where it is associated with tumor progression [162,163]. The DNA-PK inhibitor NU7026 (LY293646) has been tested for anti-cancer activity in CCA preclinical models. DNA-PK appears to be a negative regulator of DNA repair in CCA, as NU7026 treatment reduces γH2AX levels [164], suggesting that DNA-PK might exert a controversial role in CCA. Nevertheless, the orally deliverable DNA-PK inhibitor Peposertib (M3814) showed good overall tolerability and modest therapeutic response as monotherapy in patients with advanced solid tumors, including CCA [165]. This suggests that focusing on potentiating the efficacy of systemic therapy and developing drug combinations of Peposertib with chemo- and radiation approaches might be rational. Currently, NCT04068194 phase I/II study is actively recruiting patients with locally advanced or metastatic hepatobiliary malignancies to elucidate the effects of Peposertib when combined with immunotherapy (Avelumab) and/or radiotherapy [166].

#### 2.5.2. PLK1

PLK1 is the most studied member of the PLKs family of Ser/Thr protein kinases, which represents a crucial player in managing multiple aspects of cell division, including genomic stability, mitotic entry and exit and cellular response to DNA damaging insults [167,168]. PLK1 has been described as overexpressed and associated with poor prognosis in a plethora of human neoplasms, including CCA [169,170,171]. Indeed, hindering PLK1 by using blocking antibodies, genetic depletion and pharmacological molecules negatively affects cancer cell proliferation and results in the induction of apoptosis [169].

Available preclinical data suggest that the pharmacological blockade of PLK by different selective kinase inhibitors as single agents or in combination with chemo- and radiotherapeutics might offer a promising anti-cancer strategy [172,173,174,175]. Interestingly, PLK inhibition revealed favorable clinical applications for CCA patients. In fact, the phase I study NCT01348347 recently concluded with positive results about overall tolerability, safety and clinical benefits of the oral PLK inhibitor Volasertib in patients with advanced CCA [176,177].

**Table 1 cells-11-01463-t001:** Active clinical studies of DDR inhibitors in CCA patients.

Target	Treatments	Primary Endpoints	Phase	Study Identifier (ClinicalTrials.gov, accessed on 8 March 2022)
PARP	Rucaparib + Irinotecan/5-FU/Leucovorin calcium	MTD; DCR	I/II	NCT03337087 [178]
PARP	Niraparib	ORR	II	NCT03207347 [123]
PARP	Olaparib	ORR	II	NCT03212274 [112]
PARP; ATR	Olaparib + Ceralasertib (AZD6738)	ORR	II	NCT03878095 [114]
PARP; PD-L1	Olaparib + Durvalamab	ORR; DCR	II	NCT03991832 [116]
PARP	Olaparib	ORR	II	NCT04042831 [109]
PARP; PD-1	Olaparib + Pembrolizumab	ORR	II	NCT04306367 [117]
PARP; ATRATR; PD-L1	Olaparib + Ceralasertib (AZD6738) Ceralasertib (AZD6738) + Durvalamab	DCR	II	NCT04298021 [115]
PARP; PD-1	Rucaparib + Nivolumab	PFS at 4 months	II	NCT03639935 [121]
Wee1	Adavosertib (AZD1775)	ORR	II	NCT02465060 [138]
DNA-PK; PD-L1	Peposertib + Avelumab	MTD; ORR	I/II	NCT04068194 [166]

MTD: maximum dose tolerated; DCR: disease control rate; ORR: overall response rate; PFS: progression-free survival.

#### 2.5.3. Evidence and Future Prospects of DDR Inhibition-Based Combination Therapy in CCA

Several both predictive and pharmacodynamic biomarkers have been proposed for pharmacological inhibition of DDR [179]. The presence of mutations in BRCA genes undoubtedly represents the most used criteria to select cancer patients, including those with CCA, for therapeutic regimens which involve DDR inhibitors [139,179]. Recently, a genomic study in a large BTC patients cohort revealed that BRCA mutations are associated with high microsatellite instability and deficient mismatch repair, together with higher median tumor mutation burden, which is broadly known to be related to the patient’s response to immunotherapy [180,181]. BRCA status is defined as a predictive biomarker for therapeutic response to PARPi in several types of cancer, including CCA [139,181]. PARPi have been demonstrated to function as anti-tumor agents also in wild type BRCA cancers [182,183,184], suggesting that screening dysregulations in alternative DDR genes might represent a promising approach to identify other predictive biomarkers for therapeutic response to PARPi and more DDR targeting molecules.

Namely, in iCCA patients, inactivating mutations in BAP1 and amplifications in RAD21 gene provide two novel predictive biomarkers for the clinical efficacy of PARP chemical inhibition [185]. Intriguingly, BAP1 loss can be frequently detected in CCA, and several reports indicate that the chemical inhibition of histone deacetylases (HDAC), which are also involved in DNA repair process, reverses the phenotypic effects of BAP1 inactivation, promoting accumulation of damaged DNA and inducing cancer cell death [186,187,188]. HDAC inhibitors (HDACi) have shown promising preclinical results as anti-CCA molecules [189], with Entinostat undergoing clinical evaluation in combination with immunotherapy [190]. Importantly at epigenetic level, inhibitors of bromodomain and extra-terminal motif (BET) proteins have been recently developed as anticancer agents. This class of inhibitors reversibly binds BET proteins (BRD2, BRD3, BRD4, BRDT) and prevents protein–protein interaction between BET proteins and acetylated histones and transcription factors. BAP1 loss displayed the ability to increase the sensitivity of cancer cells also to BET inhibitors [191], which remarkably result in potent anti-tumor effects in CCA models, especially in combination with PARPi [103]. Hence, BAP1 mutated subgroup of CCA patients might appear as favored candidates for immunoepigenetic combinatorial strategies and for approaches combining different DDRi. Therefore, NGS profiling prior to clinical trial enrolment appears an essential prerequisite to selecting and assigning specific subpopulations of CCA patients to the appropriate therapeutic direction.

As CCA is a highly desmoplastic malignancy, the profound interconnection between cancer cells and tumor microenvironment (TME) along with the inflammation status of CCA should be discussed. Interestingly, inflammation and genomic instability are closely related in a positive feedback loop. Inflammation contributes to carcinogenesis and malignant progression by inducing DNA damage through RONS (reactive oxygen and nitrogen species); DNA damage, in turn, can intensify inflammation [192]. CCAs are classified into two main groups, depending on the type and degree of immune activation. “Hot” CCAs are generally defined by extensive T-cell (CD8^+^) infiltration, large presence of anti-tumor dendritic cells and NK cells, in addition to the enhanced production of interferon γ and granzyme B and an increased exposure of PD-1 and its ligand. Conversely, “cold” CCAs display low CD8^+^ infiltration and predominance of immunosuppressive populations, just as in tumor-associated macrophages and myeloid-derived suppressive cells [193]. As mentioned, DDR inhibitors have the potential to convert “cold” tumors to “hot” tumors, enhancing the sensitivity to immune checkpoint blockade, by promoting the exposure of neoantigens on the cancer cell surface and by activating interferon signaling [194]. Thus, selecting CCA patients based on their specific immune landscape might improve therapeutic response to immunotherapy by enhancing the inflammation status through DDR inhibition.

Finally, genotoxic agents have been shown to induce DNA damage in cancer-associated fibroblasts (CAFs), which, in turn, activate the senescence program and reorganize the TME, resulting in reduced sensitivity of cancer cells to radio- and chemotherapy [195,196,197]. Targeting DDR response in such CAFs might not be beneficial, since non-malignant cells harbor a robust machinery for DNA repair and would likely activate alternative ways to compensate for a DDR pathway inhibition. An alternative and more innovative approach might combine DDR inhibitors with antifibrotic drugs which specifically target CAFs by blocking the activity of PDGFRβ, VEGFR3, VEGFA, VEGFC, and EGFR. This strategy has shown promising results as a single-agent in preclinical settings [198], suggesting that TME represents a core element in CCA not only for preserving a pro-tumoral context, but also for providing novel possibilities for drug application. Figure 1 shows the main druggable DDR proteins within specific pharmacological inhibitors and the biological effects that have been observed in CCA cells.

## 3. Discussion

Alterations in DDR-related genes such as BRCA1/2, PARP, ATM, ATR, BAP1, ARID1A, RAD51, MLH1, TP53 makes CCA an optimal candidate for DDRi treatment, a well-established therapy in clinical application today.

However, the therapy resistance observed against DDRi in aggressive tumor types including iCCA raises the question of how to improve the treatment efficacy. Numerous phase studies show that combinatory therapy approaches using DDRi with other therapy regimens could be the next step in the translational benefit of these inhibitors.

Constituting the 80% of DDR activity in a cell, PARPi basically forms the core of DDR-targeting therapies. As mentioned in the previous section, PARPi (Olaparib, Niraparib, Veliparib, Talazoparib, Rucaparib) showed great promise in clinical studies when combined with conventional chemotherapeutics such as Gemcitabine or Cisplatin, and with immunotherapeutics. Additionally, PARPi were shown to be more effective on tumor cells with specific mutations, such as IDH1/2. Altogether, these properties make PARPi the best possible candidates for further therapeutic developments.

Along with PARPs, Wee1 is considered a useful player to target in cancer cells, since it specifically controls the G2/M checkpoint, and helps malignant cells to maintain a sustainable degree of genomic instability. Wee1 inhibitor Adavosertib was proven useful and thus this small molecule has been included in further clinical studies where it was administered alone or in combinatorial regimens with diverse therapeutics. As a result, it has been shown to be successful in both strategies, and bears hope to develop novel therapeutic approaches for iCCA patients.

Studies performed with the inhibitors developed against ATM/ATR and their downstream elements CHK1/2 remain insufficient to date. Even though there are no current ATMi clinical trials, specific ATRi are currently being evaluated by phase-II studies for therapeutic improvements after combined administration with PARPi or immuno-therapy in patients with solid CCA who have failed the first-line systemic chemotherapy. A similar picture is observed in CHK1/2-targeting studies, being scarce at this moment. However, CHKi appear promising because of their ability to sensitize malignant cells to further therapeutic applications such as radiotherapy or platin-based drug administration.

Even though the most prominent targets are the ones discussed above, other candidates have come into the light in recent years, such as DNA-PK and PLK. Because their inhibitors have exhibited positive results in preliminary studies, they represent a viable target for further research.

However, the success of a treatment is not limited to the development and use of the best inhibitor, but also to the genomic condition of the patient. In order to obtain the best outcome, this needs to be taken into elaborate consideration. Related research has shown that BRCA mutations, which are the most-known alteration in CCA cells in addition to many other solid tumors, may not be the best predictive biomarker for the accurate estimation of DDRi response in CCA patients. Therefore, better biomarker candidates must be identified. Studies have shown that BAP1-inactivating mutations and RAD21 amplifications contribute to the efficacy of PARPi, pointing to the fact that NGS profiling of tumors from patients is the best tool for determining the appropriate therapeutic strategy, especially in targeting tumor subpopulations and decreasing the recurrence possibility. Use of immunotherapy in combination with DDRi to obtain the best response from patients with specific immune characteristics must also be considered. Finally, CAFs that exert powerful effects on TME reorganization should be taken into consideration in designing DDRi therapy. Following a strategy that combines antifibrotic drugs to eliminate these cells represents a rational approach. Future innovative multi-targeted strategies focusing on CCA-intrinsic pathways and TME-extrinsic mediators will likely improve therapeutic efficacy, advancing treatment of this disease.

## Figures and Tables

**Figure 1 cells-11-01463-f001:**
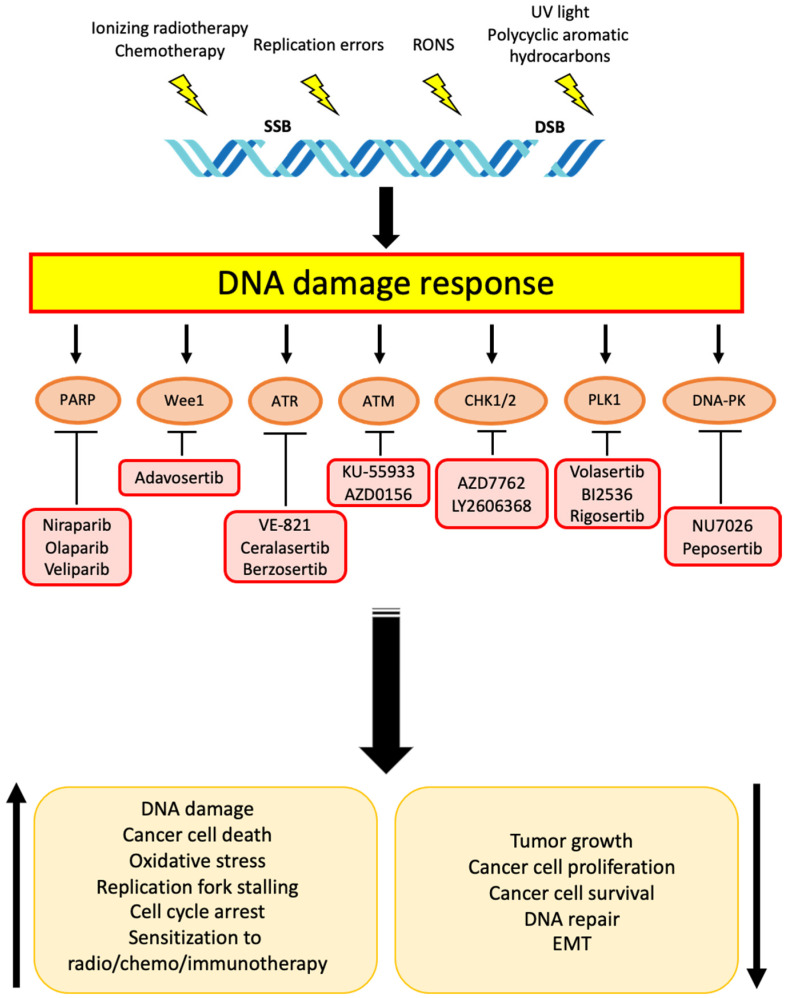
Schematic representation of DDR inhibitors used in preclinical and clinical studies for CCA therapy. Both exogenous and endogenous factors induce DNA damage (SSBs, DSBs). DNA ruptures activate DNA damage response (DDR) pathways. The main DDR proteins within their specific pharmacological inhibitors and the biological effects that have been observed in CCA cells are shown.

## Data Availability

Data sharing not applicable. No new data were created or analyzed in this work.

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
