# Peer review of "DNA Damage Response Inhibitors in Cholangiocarcinoma: Current Progress and Perspectives"

_cells, 2022, doi:10.3390/cells11091463_

Round 1
Reviewer 1 Report
In this manuscript entitled “DNA Damage Response Inhibitors in Cholangiocarcinoma: Current Progress and Perspectives” Geyik and colleagues present a comprehensive picture of the recent clinical research in Cholangiocarcinoma with DDR inhibitors. They also discuss the future perspectives of these inhibitors for the treatment of Cholangiocarcinoma, mostly in the context of combinatorial treatments.
Major comments:
- Authors do not present the DDR mutation profile of Cholangiocarcinoma. There is no reference to TCGA data or to published studies. Basic information about the frequency of BRCA1/2 or IDH1 mutations in Cholangiocarcinoma is missing.
- There is no reference or text, related to synthetic lethality concept in targeting DDR.
Minor comments:
- I suggest authors to include info for the clinical trial phase (I,II,III) in Table 1.
- I suggest the DDR factors that are related to Cell cycle progression (Wee1, CHK1, CHK2, PLK1) to be analyzed separately from DDR factors related to DNA damage repair.
Author Response
To the editor,
Regarding advised comments on our review with the title “DNA Damage Response Inhibitors in Cholangiocarcinoma: Current Progress and Perspectives”, we have revised the points given below:
Reviewer 1:
Major comments:
- Authors do not present the DDR mutation profile of Cholangiocarcinoma. There is no reference to TCGA data or to published studies.
DDR mutation profiles were already presented in the text and published studies (including TCGA data: ref.s 51, 170) were mentioned in the first paragraph of 2nd section (starting at line 101). However, additional details were integrated about these large-scale studies. Basic information about the frequency of BRCA1/2 or IDH1 mutations in Cholangiocarcinoma is missing. Related information was specified (lines 108 and 169).
- There is no reference or text, related to synthetic lethality concept in targeting DDR.
Ref.s 52, 53, 63, 76, 141, and 184 are about synthetic lethality. In addition, a detailed explanation was added on lines 119 and 123.
Minor comments:
- I suggest authors to include info for the clinical trial phase (I,II,III) in Table 1.
We have integrated the clinical phase data in Table 1.
- I suggest the DDR factors that are related to Cell cycle progression (Wee1, CHK1, CHK2, PLK1) to be analyzed separately from DDR factors related to DNA damage repair.
Authors believe that the current outline is fit for the point that is wanted to be made. Reviewer’s suggestion would hinder the primary objective of this review, which is to give a holistic approach to DDR factors and would show the DDR targets as if they are separate entities.
In addition to these points, as suggested, we also had English proofreading support (please find related document attached).
Reviewer 2 Report
Cholangiocarcinoma (CCA) is the higher lethal cancer and lacks clear biology to develop targeted therapy. In this study, the authors collect the DNA damage response and inhibitor to find a novel targeted therapy for CCA. However, Some questions need to answer.
- Radiotherapy is one of the methods to induce DNA damage responses. In this manuscript, there is a little introduction to radiation.
- H2AX is an important protein in DNA damage. However, the authors hardly discuss the role of H2AX in CCA.
- Rad51 is a kind of DNA repair protein and a homologous recombination (HR) repair pathway. If the author included Rad51 in DNA damage response. Why were Ku70 and Ku80, a Non-homologous end joining (NHEJ) pathway, excluded?
- Wee1 is a nuclear kinase belonging to the Ser/Thr family of protein kinases and induced cell cycle arrest through inhibiting CDK1. However, before Line 172, there are not any sentence about Wee1. Why did the authors introduce Wee1 in 2.2.
- In table 1, Active clinical studies of DDR inhibitors in CCA patients. The authors introduce the inhibitors of PARP, ATR, PD-L1, PD-1, Wee1, and DNA-PK. Why did the anthors refer to PD-L1 and PD-1 and exclude PLK1 inhibitor.
Author Response
To the editor,
Regarding advised comments on our review with the title “DNA Damage Response Inhibitors in Cholangiocarcinoma: Current Progress and Perspectives”, we have revised the points given below:
Reviewer 2:
- Radiotherapy is one of the methods to induce DNA damage responses. In this manuscript, there is a little introduction to radiation.
A paragraph focused on radiotherapy-related information was included starting at line 74.
- H2AX is an important protein in DNA damage. However, the authors hardly discuss the role of H2AX in CCA.
Authors have only discussed proteins related to DNA damage and DDR that are currently druggable in CCA. In that sense we believe including H2AX would digress us from the scope of this review.
- Rad51 is a kind of DNA repair protein and a homologous recombination (HR) repair pathway. If the author included Rad51 in DNA damage response. Why were Ku70 and Ku80, a Non-homologous end joining (NHEJ) pathway, excluded?
Since Rad51 has been cited only in the list of altered DDR genes in CCA, and Ku70 and Ku80 are not reported in CCA, we have only mentioned those druggable targets.
- Wee1 is a nuclear kinase belonging to the Ser/Thr family of protein kinases and induced cell cycle arrest through inhibiting CDK1. However, before Line 172, there are not any sentence about Wee1. Why did the authors introduce Wee1 in 2.2.
Actually, as for the other druggable targets, a brief introduction for each is provided in section 2.
- In table 1, Active clinical studies of DDR inhibitors in CCA patients. The authors introduce the inhibitors of PARP, ATR, PD-L1, PD-1, Wee1, and DNA-PK. Why did the anthors refer to PD-L1 and PD-1 and exclude PLK1 inhibitor.
PD-1 and PD-L1 inhibitors have been cited just because they are used in combination with DDR targeting molecules, PLK1 inhibitors are not included in the table, because that table includes only ongoing clinical trials and currently there are no active clinical studies involving PLK1 inhibitors.
In addition to these points, as suggested, we also had English proofreading support (please find related document attached).
Round 2
Reviewer 1 Report
Authors have addressed my comments and the revised manuscript has been improved.